# Endotracheal Intubation Success Rate in an Urban, Supervised, Resident-Staffed Emergency Mobile System: An 11-Year Retrospective Cohort Study

**DOI:** 10.3390/jcm9010238

**Published:** 2020-01-16

**Authors:** Michèle Chan, Christophe A. Fehlmann, Mathieu Pasquier, Laurent Suppan, Georges L. Savoldelli

**Affiliations:** 1Division of Emergency Medicine, Department of Anaesthesiology, Clinical Pharmacology, Intensive Care and Emergency Medicine, Geneva University Hospitals and Faculty of Medicine University of Geneva, CH-1211 Geneva, Switzerland; michele.chan@hcuge.ch (M.C.); laurent.suppan@hcuge.ch (L.S.); 2Emergency Department, Lausanne University Hospital, and University of Lausanne, CH-1011 Lausanne, Switzerland; 3Division of Anaesthesiology, Department of Anaesthesiology, Clinical Pharmacology, Intensive Care and Emergency Medicine, Geneva University Hospitals and Faculty of Medicine University of Geneva, CH-1211 Geneva, Switzerland; georges.savoldelli@hcuge.ch; 4Unit of Development and Research in Medical Education (UDREM), Faculty of Medicine, University of Geneva, CH-1211 Geneva, Switzerland

**Keywords:** endotracheal intubation, airway management, prehospital emergency care, emergency medical services

## Abstract

Objectives: In the prehospital setting, endotracheal intubation (ETI) is sometimes required to secure a patient’s airways. Emergency ETI in the field can be particularly challenging, and success rates differ widely depending on the provider’s training, background, and experience. Our aim was to evaluate the ETI success rate in a resident-staffed and specialist-physician-supervised emergency prehospital system. Methods: This retrospective study was conducted on data extracted from the Geneva University Hospitals’ institutional database. In this city, the prehospital emergency response system has three levels of expertise: the first is an advanced life-support ambulance staffed by two paramedics, the second is a mobile unit staffed by an advanced paramedic and a resident physician, and the third is a senior emergency physician acting as a supervisor, who can be dispatched either as backup for the resident physician or when a regular Mobile Emergency and Resuscitation unit (Service Mobile d’Urgence et de Réanimation, SMUR) is not available. For this study, records of all adult patients taken care of by a second- and/or third-level prehospital medical team between 2008 and 2018 were screened for intubation attempts. The primary outcome was the success rate of the ETI attempts. The secondary outcomes were the number of ETI attempts, the rate of ETI success at the first attempt, and the rate of ETIs performed by a supervisor. Results: A total of 3275 patients were included in the study, 55.1% of whom were in cardiac arrest. The overall ETI success rate was 96.8%, with 74.4% success at the first attempt. Supervisors oversaw 1167 ETI procedures onsite (35.6%) and performed the ETI themselves in only 488 cases (14.9%). Conclusion: A resident-staffed and specialist-physician-supervised urban emergency prehospital system can reach ETI success rates similar to those reported for a specialist-staffed system.

## 1. Introduction

### 1.1. Background and Importance

In the prehospital setting, endotracheal intubation (ETI) is often required to secure the patient’s airways [1]. An ETI is indicated in various conditions, including cardiac arrest (CA) [2,3,4,5,6], respiratory distress refractory to standard treatment [7,8], coma and traumatic injuries [9,10].

ETI in emergency situations, particularly in prehospital settings, has been shown to be more challenging than in elective situations [11]. The organization of emergency medical systems varies significantly and in many ways from one country or region to another. Similarly, the characteristics of prehospital providers performing ETI differ in many aspects [12]. It is therefore unsurprising that ETI success rates differ widely between systems, with reported values ranging from 58% to 100% in recent studies [13,14,15]. Differences in the provider’s profession (e.g., physician versus paramedic), experience, curriculum, and training contribute to this variability [13,16,17]. Other factors, such as the characteristics of the patient population studied, environmental and technical conditions, and the use of rapid sequence induction (RSI), also affect the ETI success rate. The unfavorable outcomes associated with ETI in some studies could therefore be linked to provider skill rather than to the procedure itself [13,18]. ETI success rates in systems where junior physicians are supervised by a senior expert physician in the prehospital setting have, however, scarcely been described.

### 1.2. Goals of This Investigation

The aim of our study was to evaluate the overall ETI success rate in a resident-staffed and specialist-physician-supervised emergency prehospital system. Our hypothesis was that supervised non-specialist residents would achieve an ETI success rate comparable to those reported in systems staffed only by specialist physicians. As many ETI procedures are performed because of CA and as these situations present particular challenges, a subgroup analysis of these cases was also planned.

## 2. Patients and Methods

### 2.1. Study Design and Setting

The study was conducted at the Geneva University Hospitals, Geneva, Switzerland, in accordance with Good Clinical Practice (Declaration of Helsinki 2002). This study was approved on 23 April, 2019, by the institutional ethics committee of Geneva, Switzerland (Project ID 2019-00679). Patient consent was waived by this committee.

Each year in the city of Geneva, the medical mobile unit referred to as SMUR (Service Mobile d’Urgence et de Réanimation, i.e., Mobile Emergency and Resuscitation Service) performs approximately 5000 missions and, as the only emergency medical mobile unit in Geneva, takes care of all prehospital life-threatening emergencies in the area. Medical records are completed by the lead prehospital physician. For quality and teaching purposes, all records are reviewed daily by a supervisor, corrected if necessary, and entered into a database. A retrospective chart search and review of all patients requiring prehospital ETI was performed on this database.

The emergency prehospital response system in Geneva consists of three levels of increasing expertise. The first component is an advanced life-support (ALS) ambulance staffed by two paramedics. If dispatchers deem it necessary or if the first-responding paramedics request it, the SMUR unit is engaged. This unit is staffed by a paramedic and a resident physician, and the specialization and level of experience of the latter is variable. The third and highest level relies on a specialist physician who acts as a supervisor and who can be dispatched at any time, either as backup for the resident physician (for teaching or supervision purposes) or when a regular SMUR unit is not available. Supervisors are physicians certified for prehospital emergency medicine and work full-time in the prehospital emergency unit.

Resident physicians must have at least three years of post-graduate clinical practice and work in the SMUR unit for periods of three to four months. Despite mandatory basic training, ETI skills vary significantly from one resident to another, as they come from different backgrounds.

RSI is systematically performed to facilitate ETI for patients who require it and who are not in CA. Supervisors oversee residents performing the procedure until the resident is deemed autonomous. When dealing with a patient in CA, all physicians, regardless of their background, are allowed only one attempt at intubation before they must call the supervisor. As the metropolitan area of Geneva is relatively small (less than 16 km^2^) and the time between the dispatch and the arrival of professional rescuers is correspondingly short, supervisors can quickly reinforce residents directly onsite. Paramedics are not allowed to intubate in this system.

### 2.2. Patient Population

This retrospective cohort study included all patients 18 years old or over for whom ETI was attempted from 1 January, 2008 to 31 December, 2018. Patients already intubated before the arrival of the SMUR were excluded.

### 2.3. Data Collection

Data from patient prehospital records were electronically extracted. When data were missing, the authors manually browsed the records to complete the extraction, if possible. In cases where data could not be found manually, the information was considered as definitely missing and presented as such.

Based on the different supervision rules regarding ETI in our emergency response system, patients were classified into two groups according to their CA status. Patients in the “CA” category had experienced CA before advanced airway management (induction of general anesthesia and/or ETI). When CA followed the induction procedure, patients were categorized as “without CA” and the cause of CA was described. To study the relationship between advanced airway management and the occurrence of CA, three authors with a prehospital background who do not act as SMUR supervisors (C.F., M.P., and G.S.) independently reviewed the medical records (including the history, physical status, choice of medication, and description of the intervention) to specifically determine the appropriateness of the ETI indication and the presence of a potential causality link between airway management (the induction of general anesthesia and/or ETI) and CA. In cases where judgment was difficult, disagreements were discussed by the three authors to reach a consensus.

The primary pathology described the category of the main clinical diagnosis, i.e., trauma (blunt or penetrating trauma, drowning, traumatic brain injury, hanging, etc.) versus medical (sepsis, acute respiratory failure, intoxication, non-trauma coma, etc.). Night interventions were defined as those occurring between 19:00 and 07:00. “Senior-physician interventions” were those realized directly by a supervisor on call (acting as physician first responder) with no other SMUR physician present. “Supervised interventions” were interventions during which the SMUR resident physician was supported by a specialist physician acting as a supervisor. Supervision was provided either remotely by phone call (e.g., for advice on patient management) or onsite (e.g., when support for intubation was required). The length of the intervention was the time elapsed between the unit’s dispatch and the arrival at hospital. The number and type (remote or onsite) of supervisor interventions were also recorded.

### 2.4. Outcomes

The primary outcome was the global ETI success rate. The correct placement of the tracheal tube had to be confirmed by capnography, which has been systematically used in our system since 2003 [19]. The secondary outcomes were the rate of ETI success at the first attempt, the number of ETI attempts, and the rate of ETIs performed by a supervisor. An attempt was defined as the insertion of the laryngoscope blade into the oropharynx whether or not an attempt was made to insert the endotracheal tube [20]. The definition of how an ETI attempt was to be coded was shared by all the supervisors in charge of the charts’ correction and validation and taught to both the residents and the paramedics during their initial training and continuing educational sessions. The original data were deposited to Mendeley Data [21].

### 2.5. Statistical Analysis

The Wilcoxon–Mann–Whitney test and a chi-squared test were used for group comparisons when appropriate. A two-sided *p*-value below 0.05 was considered significant. Demographic data are expressed as means ± standard deviation. Results are expressed as means or proportions and 95% confidence intervals (95 CI). Subgroup analyses were realized based on the CA status, trauma status, and provider status (supervisor or not). A sensitivity analysis was performed after the exclusion of patients for whom resuscitation attempts were quickly withheld for either medical and/or ethical reasons. Statistical analysis was performed using STATA version 15 (Stata Corporation, College Station, TX, USA).

## 3. Results

### 3.1. Characteristics of Study Subjects

A total of 47,876 adult patients were taken care of by the SMUR from 1 January, 2008 to 31 December, 2018, and their charts were examined for eligibility. We initially included 3283 patients and subsequently excluded eight of these: seven who had been intubated before the arrival of the SMUR and one duplicate file. Finally, 3275 records (6.8%) were included in the analysis (Figure 1).

Table 1 summarizes patient characteristics. Most patients were men, with a mean age of 62 ± 20 years, and presented a medical (non-trauma) primary pathology (82.4%). More than half of the patients (55.1%) were in CA before the arrival of the SMUR or before airway management. For the 42 patients in whom CA occurred during or after airway management, it was determined that the occurrence of CA was related to the airway management in 21 of these cases and unavoidable in eight cases. For all 21 patients, ETI was considered either very likely indicated or definitely indicated.

### 3.2. Main Results

The global ETI success rate was 96.8% (3170 successful ETIs). The ETI success rate was higher for patients without CA compared to those with CA (98.9% (98.2–99.4) and 95.1% (94.0–96.1), respectively; *p* < 0.001) (Table 2). After the exclusion of 56 patients for whom resuscitation attempts were quickly withheld, the global ETI success rate was 98.5% with no significant difference remaining between patients without and with CA (98.9% (98.2–99.4) and 98.1% (97.4–98.7), respectively; *p* = 0.065).

There was no statistically significant difference in the global ETI success rate between trauma and non-trauma patients (97.6% (96.0–98.7) and 96.6% (95.9–97.3), respectively; *p* = 0.244) or between intervention by a supervisor or by residents (98.3% (96.9–99.3) and 96.6% (95.8–97.2), respectively; *p* = 0.058), although it could be clinically relevant. The detailed characteristics of the 105 patients with failed intubation are shown in Table 3.

The ETI success rate at the first attempt was higher in patients without CA, even after the exclusion of patients with failed intubation (78.1% (75.9–80.2) and 72.8% (70.6–74.9), respectively; *p* < 0.001). ETI success at the first attempt was also higher in interventions performed by supervisors than for those performed by residents (89.5% (86.0–92.3) and 72.2% (70.5–73.8), respectively; *p* = 0.001). There was however no significant difference between trauma and non-trauma patients (72.4% (68.5–76.0) and 74.8% (73.1–76.4), respectively; *p* = 0.229). Finally, ETIs by supervisors were more frequent for patients without CA than those with CA (17.1% (15.3–19.1) and 13.1% (11.6–14), respectively; *p* = 0.011) as well as for patients with trauma with CA compared to those without CA (25.5% (22.1–29) and 12.6% (11.4–13), respectively; *p* < 0.001).

When supervisors were onsite with a resident physician, they performed the ETI in 41.8% of the cases. In these situations, success rates were similar regardless of patient conditions.

## 4. Discussion

The results of this study indicate that, in this resident-staffed and specialist-physician-supervised urban emergency prehospital system, the global success rate of emergency ETIs over 11 years was 96.8% and reached 98.5% after the exclusion of patients for whom resuscitation attempts were quickly withheld. Success at the first attempt was 74.4%. For patients without CA, both the global success rate and the success at the first attempt were even higher (98.9% and 77.5%, respectively).

These results suggest that a resident-staffed, supervised prehospital emergency medical system is able to provide ETI success rates similar to those of systems staffed only by senior specialist physicians and could therefore be acceptable in systems where it is impossible to have only specialized physicians, be it for financial reasons or for human-resource issues. Indeed, a recent meta-analysis found a global success rate of 98% in such systems (95% CI 97–99%) [13]. Although some studies reported success rates of up to 100% [22,23], the sample size and the duration of observation of these studies were generally limited. Moreover, the patient populations described in these studies were often chosen for a particular medical condition. Specific populations and medical problems can, however, be associated with different ETI success rates. For example, cervical spine immobilization for trauma patients, achieved either manually or when using a cervical collar, has clearly been associated with more difficult ETI [24]. Therefore, aiming for a global success rate of 100% in all situations over a long period of time for prehospital settings is probably unrealistic.

We observed that physician supervisors were more frequently present onsite when induction was required for patients without CA and for trauma patients. As patients who are not in CA are more prone to RSI-related complications (such as hypoxemia and hypotension), supervisors were therefore more often called upon in these situations [25,26,27]. Supervisors were also more frequently dispatched to deal with trauma patients, as both cervical immobilization and facial traumas are risk factors for difficult ETI [24]. Despite the presence of a supervisor, ETI was performed by residents in more than half of the cases. Onsite teaching is an extremely important aspect of this system, and this finding reflects this emphasis. Direct observation by supervisors also allows non-specialist physicians to perform ETI autonomously once they have been deemed to have enough experience.

The analysis showed that the ETI success rate was lower for CA patients. In many cases involving patients with CA, the decision to withdraw resuscitation attempts was made. When attending a patient with CA, the first priority of prehospital providers is to begin life-support measures. Data regarding advanced care planning, pre-existing medical conditions, and the CA context are gathered during the team response to facilitate the decision to either continue the resuscitation or to withdraw resuscitation efforts. ETIs which would have eventually succeeded in those patients, given one or more additional attempts, were therefore classified as failures. In fact, the ETI success rate for CA patients in our study is higher than many reported in the literature for this patient population. Moreover, as the out-of-hospital CA survival rate in Geneva is in the higher end of the range, it seems unlikely that the withdrawal of resuscitation efforts in these cases resulted from failed ETI attempts [28].

In patients with CA, the lower ETI success rate at the first attempt is probably linked to several factors. Firstly, due to time pressure, residents may not take enough time to properly prepare themselves and the patient, including correct positioning [29,30]. Secondly, all physicians and paramedics working in Geneva are instructed to continue chest compression while ETI is attempted. The neck and head movements resulting from the compression can make it more difficult to perform ETI. Thirdly, a LUCAS-2 automatic chest compression system has been used since 2010 for all non-trauma CA cases. This device requires a plate to be placed under the patient’s shoulders, which results in hyperextension of the neck that can contribute to a more difficult patient position [29]. Lastly, as residents who have not been deemed proficient to perform induction and intubation on their own are allowed to attempt ETI once for CA patients before they are required to call for help, supervisors are often called relatively late in these situations and sometimes arrive after the decision to withdraw resuscitation attempts has already been made.

This study has some limitations and strengths that should be acknowledged. Firstly, it is a retrospective study for a specific type of system (i.e., small urban area and specialist-supervised). Generalization of the results to other types of systems should therefore be considered with caution. Another limitation is the non-recorded variables in the database entries. It would indeed have been interesting to report the Cormack score, the BMI, difficult intubation criteria, peri-procedural measures (such as length of procedure or blood pressure), the duration of the ETI procedure, the etiology of the ETI failure, airway trauma and complications, and the hospital data (such as emergency department (ED) length of stay or mortality rate). Moreover, given the data collected in our charts, we were unable to assess immediate or delayed ETI-related complications. However, it seems unlikely that our analysis and our main results would have been affected by the availability of such data. Costs were also not analyzed, although it seems obvious that a system involving only residents would be less expensive than one based on senior specialists. Although our system requires an on-call supervisor twenty-four hours a day, seven days a week, these supervisors also perform other tasks, such as acting as chief medical officers and consultants to police tactical intervention units. The presence of supervisors would therefore still be required even if SMUR physicians were all specialists. Finally, most of the outcomes were self-reported, which, even if the definitions were clearly stated to everyone, could have biased the results.

We believe that the main strength of our study is the organization of the system that was studied. The emergency prehospital response system in our city is quite unique with its three levels of increasing expertise as described above. To our knowledge, this is the first study to investigate a prehospital emergency medical system staffed by residents of various levels of training and background and overseen by prehospital emergency-medicine-certified physicians who act as supervisors and are available at all times. Other strengths are the large sample size and the duration of the cohort study (48,000 clinical records recorded over 11 years, with more than 3200 patients included). Finally, the high quality of reported data allowed us to give extremely precise information (such as number of attempts, alternative device utilization, and CA occurrence) with extremely low levels of missing data.

## 5. Conclusions

In summary, this study demonstrates that a supervised, resident-staffed urban prehospital medical system can reach ETI success rates similar to those reported in specialist-staffed systems. Further studies in similar systems should confirm these findings. In the future, the cost effectiveness of such supervision should also be assessed.

## Figures and Tables

**Figure 1 jcm-09-00238-f001:**
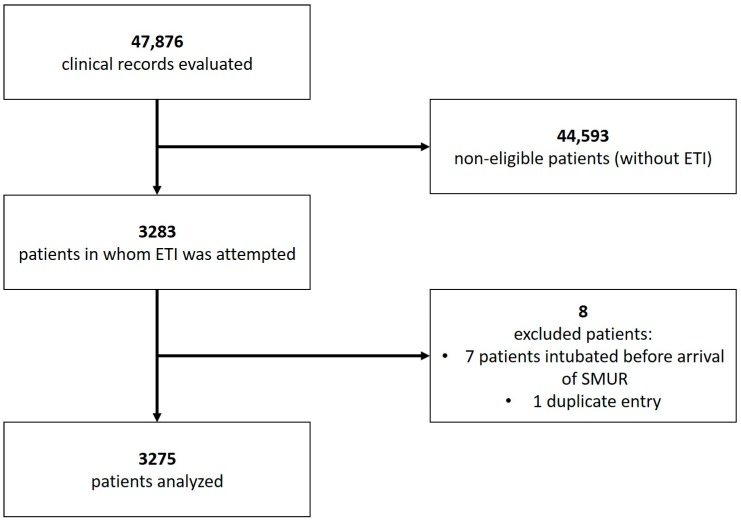
Flowchart of the study.

**Table 1 jcm-09-00238-t001:** Patient characteristics ^1^.

	All Patients (*n* = 3275)
Age (year)	62 ± 20
Sex (F)—n (%)	1182 (36.1)
Place of intervention—n (%)	
Home	2040 (62.3)
Public place	1013 (30.9)
Care structure	222 (6.8)
Primary pathology—n (%)	
Non-trauma	2699 (82.4)
Trauma	576 (17.6)
Weekend intervention—n (%)	938 (28.6)
Night intervention—n (%)	1320 (40.3)
Intervention by supervisor—n (%)	417 (12.7)
Supervised intervention—n (%)	
None	1260 (38.5)
By phone	431 (13.2)
Onsite	1167 (35.6)
Not applicable ^2^	417 (12.7)
Length of intervention (min)	64 ± 22
Cardiac arrest—n (%)	1846 (56.4)
Occurrence of cardiac arrest—n (%)	1724 (52.6)
Before arrival	80 (2.4)
After arrival, before induction	1 (0.03)
After induction, before 1st laryngoscopy	4 (0.1)
After 1st laryngoscopy, before intubation	18 (06)
After intubation, less than 5 min	18 (0.6)
After intubation, more than 5 min	1 (0.03)
Missing	

^1^ Plus–minus values are means ± standard deviation. Percentages may not total 100 due to rounding. ^2^ Not applicable indicates, in this context, that the intervention had already been realized by the supervisor on call.

**Table 2 jcm-09-00238-t002:** Primary and secondary outcomes ^1^.

	All Patients (*n* = 3275)
Intubation success—n (%)	3170 (96.8)
Success at first attempt—n (%)	2435 (74.4)
Number of attempts—n (%)	
1	2435 (74.4)
2	618 (18.9)
3	170 (5.2)
>3	41 (1.3)
Missing	11 (0.3)
Intubation by supervisor—n (%)	
No	2272 (69.4)
Yes	488 (14.9)
Not applicable ^2^	515 (15.7)

^1^ Percentages may not total 100 due to rounding. ^2^ Not applicable indicates that either the intervention was realized by the supervisor on call or that the intubation failed.

**Table 3 jcm-09-00238-t003:** Description of patients for whom intubation failed (*n* = 105).

	Patients without CA (*n* = 16)	Patients with CA (*n* = 89)
Device used for oxygenation—n (%)		
Laryngeal mask airway	10 (63)	72 (81)
Bag mask	3 (19)	12 (13)
Cricothyroidotomy	1 (6)	3 (3)
Return of spontaneous breathing	2 (13)	0 (0)
Oxygenating bougie	0 (0)	1 (1)
Oxygenation impossible	0 (0)	1 (1)
Adverse events		
Cardiac arrest	1 (6)	NA
Hypoxia	7 (44)	NA
Aspiration	2 (13)	23 (26)
Dental lesion	1 (6)	2 (2)
Indication of tracheal intubation		
Cardiac arrest	NA	89 (100)
Medical—intoxication	1 (6)	NA
Medical—respiratory distress	3 (19)	NA
Neurology	5 (31)	NA
Trauma—head injury	7 (44)	NA
Rapid sequence induction—n (%)	16 (100)	3 (3)
Early decision to withdraw resuscitation attempts—n (%) ^1^	0 (0)	56 (63)
Unsupervised	0 (0)	26 (46)
Distant supervision	0 (0)	17 (30)
Onsite supervision	0 (0)	12 (21)
Not applicable ^2^	0 (0)	1 (2)
Transport to the ED—n (%)	16 (100)	10 (11)
Attempted intubation in the ED—n (%) ^3^		
No	2 (13)	2 (20)
Yes	13 (81)	7 (70)
Missing	1 (6)	1 (10)
Successful intubation in the ED—n (%) ^4^		
No	1 (8)	2 (29)
Yes	12 (92)	5 (71)
Device successfully used—n (%) ^5^		
Macintosh laryngoscope	6 (50)	3 (60)
Fibroscopy	1 (8)	0 (0)
Intubating LMA	1 (8)	0 (0)
Airtraq	0 (0)	1 (20)
Missing	4 (33)	1 (20)

^1^ Overall percentage is for patients with cardiac arrest (CA), and categorical percentages are for patients with early decision to withdraw resuscitation attempts. ^2^ Not applicable indicates that the intervention was already realized by the supervisor. ^3^ Percentage of patients transported to the emergency department (ED). ^4^ Percentage of attempted intubations. ^5^ Percentage of successful intubations.

## Data Availability

The data that support the findings of this study are available on Mendeley Data.

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
