# Peer review of "Endotracheal Intubation Success Rate in an Urban, Supervised, Resident-Staffed Emergency Mobile System: An 11-Year Retrospective Cohort Study"

_jcm, 2020, doi:10.3390/jcm9010238_

Round 1
Reviewer 1 Report
The manuscript by Chan, et al. reports the results of a retrospective cohort of 3275 patients undergoing pre-hospital endotracheal intubation by a specific EMS system in Geneva Switzerland. The authors describe the structure of their EMS system which includes three tiers: 1) a paramedic EMS unit (which does not perform intubations), 2) a paramedic/resident EMS unit with support from an offsite specialist, and 3) a specialist-staffed EMS unit. They report an overall rate of ETI success of 96.8%, and a rate of first pass success of 74.4% which they report as comparable with success rates of EMS systems using only specialist-staffed EMS.
There are considerable strengths to the analysis including an important question (how best to intubate critically ill patients in the pre-hospital setting), and a large cohort of critically ill patients with a large amount of prospectively collected data and little missing data. While the model described in this paper is very specific to other cities with (or considering) a similar model and would not generalize to models used in many other parts of the world (such as the United States which uses only paramedic led EMS teams), there is clear value to reporting the researchers experience using this system. There are, however, several weaknesses inherent to the observational, retrospective nature of the analysis including that limit comparisons between patients intubated by the paramedic/resident EMS unit and the specialist-staffed EMS unit such as biased cohorts, self-reported outcomes, lack of available data on other important measures of successful tracheal intubation (oxygen saturations, blood pressures, duration of intubation procedure), and lack of data on and data beyond hospital arrival. Some of these problems are not modifiable, but addressing the concerns below would strengthen the analysis.
MAJOR:
The way the results are presented distracts from the primary focus of the paper. Based on the introduction and conclusions, the authors primary message would seem to be that a resident-staffed and specialist supervised model performs similarly to a specialist-staffed model, but the results all compare (using statistical testing) patients with and without cardiac arrest prior to induction. Pre-existing cardiac arrest has been repeatedly demonstrated to be a predictor of difficult intubations for all of the reasons discussed by the authors, and re-demonstrating that does not appear to be the focus of the analysis. So why, then, do the authors choose to display all results as “overall, with cardiac arrest, without cardiac arrest”? In an observational study it is difficult to compare the EMS model used in this study with other theoretical models (like a system that uses only specialist-staffed teams), but intubations performed by specialist would seem to be a logical group for comparison. I suggest that all tables be re-written as “Overall,” “Intubations first attempted by residents,” and “Intubations first attempted by specialists.” As the authors acknowledge, specialists are more likely to be called as first operator for cases/intubations that are expected to be more difficult. While this is still the closest proxy for comparing how the system in Geneva would operator if all EMS units were led by specialists, the currently reported unadjusted, univariate analyses are not ideal. Would consider reporting a multivariable, logistic regression model comparing for likely predictors of difficult intubations (cardiac arrest, trauma, etc) to estimate whether or not being intubated by a resident-staffed team is an “independent” predictor of failure (overall and on the first attempt). Alternatively, the authors could scale back the goals of the analysis to simply reporting their experiences with their system and remove any between-group comparisons, reporting only their “overall” results. Given the observational nature of the study, all outcomes are hypothesis-generating, but the authors highlight ETI success as the “primary” outcome. As reported by the authors and found in this study, successful ETI occurs in nearly all patients and may not be the most sensitive outcome for reporting quality of airway management. It misses many important complications of ETI including hypoxemia, hypotension, airway trauma. Of the outcomes available to the authors, intubation on first attempt might be considered as a superior “primary” outcome. The authors should add to the paragraph listing limitations, the following additional limitations: biased cohorts (if comparing resident and specialist led teams), self-reported outcomes, lack of available data on oxygen saturations, blood pressures, duration of intubation.
MINOR:
ABSTRACT: Page 1, line 25. Would re-phrase how the “3rd tier” (specialist unit) is described. Confusing at present (explanation in introduction is much clearer) Would suggest using non-parametric testing (Mann Whitney U) for comparing linear variables (opposed to t-test) Not clear how the researchers defined airway management as “avoidable, very likely indicated, or definitely indicated.” Also not clear how researchers defined CA as “related to airway management or not.” Table 1, page 5: numbers in “overall” column for occurrence of cardiac arrest are not correct (1724 patients overall had cardiac arrest which would be 52.6% of the total, but is listed as 93.4). Similar problems throughout that section. Instead of reporting “intubation by supervisor”; would report “2nd proceduralist required” to differentiate cases where the specialist was the first operator (tier 3 intubation) and where the specialist was the supervisor (tier 2 intubation) Discussion (page 7, line 207): “Patients requiring RSI are indeed at increased risk of complications.” This sentence is at odds with a sentence from the Methods section implies that “RSI is systematically performed.” Please clarify and report the number of patients for whom RSI was performed, in each group, if available. All data is self-reported (a clear weakness). The outcomes section suggests that a universal definition for “attempts” was used, which is good. Can the authors describe how providers were trained on these definitions?
Reviewer 2 Report
Dear Editor,
Thank you for the opportunity to read this original retrospective paper entitle: Endotracheal Intubation Success Rate in an Urban, Supervised, Resident-staffed Emergency Mobile System: an 11-year Retrospective Cohort Study, by Chan M, Fehlmann CA et al.
- The study hypothesis was that "supervised non-specialist resident" would achieve an ETI success rate comparable to those reported in systems staffed by "specialist physicians" only.
- The aim of the study was "to evaluate the ETI success rate" in a "resident-staffed" and "physician specialist"-supervised the emergency prehospital system.
The study is well written, with an average length and valuable reference.
MY-MAJOR COMMENT AND POINT OF VIEW:
M&M, the study luck of power analysis and show a statistical significant tendency between the two study groups: the difference in global ETI success rate between intervention by supervisor or by residents 98.3% - (1.7 patients) – vs 96.6% - (3.4 patients), p 0.058. In other words, there is a double failure in the group of residents.
The population studied is just an ADULT and not PEDIATRIC, and you know that pediatric intubation is another story.
- Discussion: line 196 – 198 page 7 "These results suggest that a resident-staffed supervised prehospital emergency medical system can provide ETI success rates similar to those of systems staffed only by senior specialist physicians", need to be re-write.
Replay: If your country is playing the Word-Champion Cap would you be represented by the professional team of by the junior team?
Line 231 -232 page 8 "This device requires a plate to be placed under the patient's shoulders, which results in hyperextension of the neck that can contribute to a more difficult patient positioning(29)".
Replay: Are you sure of this affirmation? This position is the preferred one by much intensive care physician and pulmonology for tracheotomy and rigid bronchoscopy.
Line 136, page 3, you are talking about ethical reasons may be. But line 245 – 246 of the discussion you say that: "Costs were also not analysed, although it seems obvious that a system involving only residents would be less expensive than one based on senior specialists". Would your parents resuscitate by an experienced physician or a junior resident without full accreditation? This mean: assurance issue and ethic issue
CONCLUSION: line 263 – 264 "In the future, the cost-effectiveness of such supervision should also be assessed.
Replay: Yes, but medicine is the analysis of the 4 "E": effectiveness, efficacy, economic and ethic. Please try to be more inclusive.
Best Regards,
